# Fibroblasts as Modulators of Local and Systemic Cancer Metabolism

**DOI:** 10.3390/cancers11050619

**Published:** 2019-05-03

**Authors:** Hannah Sanford-Crane, Jaime Abrego, Mara H. Sherman

**Affiliations:** 1Department of Cell, Developmental & Cancer Biology, Oregon Health & Science University, Portland, OR 97201, USA; sanfordc@ohsu.edu (H.S.-C.); abrego@ohsu.edu (J.A.); 2Knight Cancer Institute, Oregon Health & Science University, Portland, OR 97201, USA

**Keywords:** cancer-associated fibroblast, cancer metabolism, tumor-stroma crosstalk

## Abstract

Fibroblast activation is an accompanying feature of solid tumor progression, resembling a conserved host response to tissue damage. Cancer-associated fibroblasts (CAFs) comprise a heterogeneous and plastic population with increasingly appreciated roles in tumor growth, metastatic capacity, and response to therapy. Classical features of fibroblasts in a wound-healing response, including profound extracellular matrix production and cytokine release, are recapitulated in cancer. Emerging evidence suggests that fibroblastic cells in the microenvironments of solid tumors also critically modulate cellular metabolism in the neoplastic compartment through mechanisms including paracrine transfer of metabolites or non-cell-autonomous regulation of metabolic signaling pathways. These metabolic functions may represent common mechanisms by which fibroblasts stimulate growth of the regenerating epithelium during a wound-healing reaction, or may reflect unique co-evolution of cancer cells and surrounding stroma within the tumor microenvironment. Here we review the recent literature supporting an important role for CAFs in regulation of cancer cell metabolism, and relevant pathways that may serve as targets for therapeutic intervention.

## 1. Introduction

The increasingly appreciated role of activated fibroblasts in cancer progression and response to therapy [1] has prompted investigation of growth-permissive fibroblast functions in cancer. Cancer-associated fibroblasts (CAFs) derived from activated resident fibroblast pools or from mesenchymal progenitors recruited to the tumor microenvironment [2,3,4] have displayed diverse pro-tumorigenic functions that cooperate with cell-autonomous mechanisms to promote the hallmarks of cancer [5]. While CAF functional diversity is increasingly appreciated across tissue sites [6,7,8] and precise functions may vary in a tissue-specific manner, CAFs have conserved and well-documented roles in establishing key components of a wound-healing reaction in solid tumor tissues. While fibroblastic cells evolved to play roles in tissue homeostasis and wound healing, CAFs reflect both the classical role of fibroblasts in tissue biology and unique roles resulting from co-evolution with neoplastic growths. CAFs are typically abundant in solid tumors [9], and are the principal producers of extracellular matrix (ECM) components and remodeling enzymes [10]. In addition, CAFs secrete numerous signaling proteins including mitogenic growth factors that can stimulate proliferation in the epithelial compartment [1], as well as pro-inflammatory mediators that can modulate intratumoral immune infiltration [7,11,12,13]. These established CAF functions resemble those of activated fibroblasts to support regeneration and repair. More recently, however, evidence has emerged to support a critical role for CAFs as regulators of critical metabolic processes in cancer [14]. These metabolic roles may be specific to fibroblastic cells in tumor microenvironments, as adaptive mechanisms to support the metabolic demands of rapidly proliferating cancer cells. Supporting this connection, recent analysis of metabolic networks in human breast cancer in situ showed significant correlation of intracellular metabolic states of cancer cells and adjacent CAFs [15], and mechanistic studies are beginning to uncover the importance of this bioenergetic coupling of tumor and stroma. Below, we discuss the emerging roles of CAFs in regulation of cellular and organismal metabolism in cancer.

## 2. Wound-Healing Mediators as Metabolic Regulators

As critical regulators of the wound-healing reaction, activated fibroblasts are key producers of soluble secreted factors such as cytokines, growth factors, and ECM components that orchestrate wounding-associated inflammation, regeneration, and tissue repair. CAFs similarly produce classical wound-healing mediators, and emerging evidence suggests that beyond their long-appreciated roles in repair processes, these factors also regulate metabolic functions of neighboring cancer cells. Among solid tumors, pancreatic ductal adenocarcinoma (PDAC) has a particularly prominent stromal compartment, characterized in part by a prominent CAF population and a dense, collagen-rich ECM [16]. This ECM has been shown to restrict vascular perfusion in the PDAC tumor microenvironment (TME) [17,18], and perhaps due to low serum availability, the PDAC milieu is nutrient-poor [19]. Under conditions of low nutrients, mutant KRAS, a main oncogene of PDAC, is able to drive macropinocytosis as a source of amino acids from extracellular proteins for the cancer cells [20]. Recent work has shown that the abundant, CAF-derived collagen in the PDAC ECM can serve as an extracellular protein source, and one particularly rich in prolines (25% of the amino acids in collagen) [21]. PDAC cells can take up collagen fragments, through macropinocytosis-dependent and -independent mechanisms, and subsequently metabolize proline via proline oxidase (POX/PRODH1) to fuel the TCA cycle and promote proliferation and survival under nutrient-restricted conditions in vitro or during tumor growth in vivo. Consistent with this study, soluble CAF-derived proteins together with 3D type I collagen induce transcriptional and metabolic alterations in PDAC cells supporting anabolic programs, which overlaps significantly with networks regulated by oncogenic KRAS and suggests points of convergence between cell-intrinsic and microenvironmental mechanisms that regulate cancer cell metabolism [22]. CAF-derived cytokines including CCL5, IL6, and CXCL10 can also regulate cancer cell metabolism by promoting phosphorylation of phosphoglucomutase 1 and increasing glycogen mobilization in cancer cells, promoting NADPH synthesis and the TCA cycle and thus enabling cancer cell proliferation and metastatic spread of ovarian cancer cells [23] (Figure 1).

Beyond uptake, ECM components produced by CAFs also regulate cancer cell metabolism via activation of diverse signaling mechanisms. In addition to collagens, CAFs also produce and secrete high levels of hyaluronan (HA), as well as enzymes that break down and remodel the ECM. A recent study demonstrated that HA fragments can signal through receptor tyrosine kinases to induce ZFP36, causing degradation of *TXNIP* transcripts and subsequently blocking TXNIP-mediated internalization of glucose transporter GLUT1 [24]. This leads to an increase of GLUT1 transporter on the plasma membrane, increasing the amount of glucose transport, and inducing glycolysis within the cancer cell. ECM signals act on TXNIP for acute and protracted regulation of glucose uptake, showing that external cues can regulate cellular metabolism and migration. Increased ECM stiffness during tumor progression and downstream mechanosensing induces CAFs to release aspartate, supporting cancer cell proliferation, while cancer cells in turn secrete glutamate and balance the redox state of CAFs to further promote ECM remodeling [25]. A stiff ECM mechanoactivates the YAP/TAZ pathway which plays a central role in cell proliferation, survival, and polarity, especially in tumor cells. Mechanostimuli of the ECM is thus linked to tumor cell metabolism, while tumor cell metabolism is linked to responses by the CAFs to increase ECM stiffness, resulting in a positive feedback between CAFs and cancer cells. While ECM stiffness and poor perfusion can reduce drug delivery and promote chemoresistance, CAFs can also promote chemoresistance through the release of glutathione and cysteine [26]. Glutathione and cysteine are released by CAFs leading to increased GSH levels in cancer cells, and to a reduction of platinum accumulation in cells treated with platinum-based therapies. Interestingly, CD8 T cells reverse this chemoresistance mechanism through release of interferon-gamma, which causes upregulation of gamma-glutamyltransferase activity in CAFs and to transcriptional repression of system xc^-^ cystine and glutamate antiporter via JAK/STAT signaling.

CAFs regulate the anti-tumor immune response through secretion of numerous immunomodulatory factors (reviewed in reference [27]). Fibroblasts secrete similar factors as part of the wound-healing response to recruit immune factors to an injury, however during cancer progression CAF secreted factors generally have an immune-suppressive function. The immune cells regulated by CAFs can in turn impact cancer cell metabolism, highlighting the complexity of metabolic regulation within an intact TME. CAF secretion of CXCL12/SDF1, M-CSF/CSF-1, IL-6, and CCL2/MCP-1 recruits tumor-associated macrophages (TAM) to the TME and actively differentiates TAMs into an M2 immunosuppressive phenotype. In addition, CAF secretion of CXCL1, CXCL2, CXCL5, CXCL6, CXCL8, and CCL2 recruits tumor-associated neutrophils (TANs) to the TME and polarizes them to an N2 pro-tumoral phenotype. TGF-β, secreted by CAFs, induces miR-183 to inhibit DAP12 transcription and results in reduced natural killer (NK) activating receptors (NKp30, NKp44, NKG2D) on the NK cell surface. Along with its impact on NK cells, TGF-β also causes dendritic cells (DC) to downregulate MHC class II expression, along with CD40, CD80, and CD86 leading to decreased antigen presentation efficiency and decreased production of TNF-α, IFN-γ, and IL-12, ultimately causing a reduction in T cell recruitment and survival in the TME. PGE2 and IDO secretion by CAFs affects NK cells by decreasing their cytotoxicity against cancer cells [28]. In lung cancer, TDO2 secretion by CAFs promotes tryptophan metabolism to kynurenines (Kyn), inhibiting DC differentiation while VEGF secretion inhibits DC generation and maturation by reducing MHC class II expression and antigen presenting abilities [29]. TGF-β promotes cell death of CD8+ T cells by inhibiting expression of the pro-survival factor Bcl-2. IDO1 secretion further damages T cell response by catabolizing tryptophan degradation into Kyn, creating an immunosuppressive TME and causing T cell anergy and apoptosis through depletion of tryptophan combined with an accumulation of immunosuppressive tryptophan catabolites. CD4+ helper T lymphocytes react to CAF secretion of CCL2, CCL5, and CCL17 along with polarizing cytokines IL-1, IL-6, IL-13, and IL-26 by switching from an anti-tumor T_H_1 response to a pro-tumor T_H_2 and T_H_17 response. CAFs secrete immunomodulatory factors that regulate the immune response within the tumor niche by creating an immunosuppressive environment which decreases the antigen presenting capabilities of NKs and DCs while simultaneously decreasing cytotoxicity and survival of T cells. Together, these immunomodulatory functions of CAFs can profoundly impact cancer metabolism, and regulate cancer progression through both immune effector and metabolic mechanisms.

Aging adds another variable to the impact of the TME on cancer development. The effect of age on the TME is so significant that genetically identical cells can have varying metastasis levels and therapy response based on whether they are in an aged TME or young TME, suggesting that the aged TME is capable of profoundly influencing cancer cell behavior [30]. Indeed, aged fibroblasts secrete sFRP2, a Wnt antagonist, which causes a downstream cascade of signals and subsequent loss of β-catenin and MITF, as well as redox effector APE1. This loss of APE1 makes melanoma cells less responsive to ROS-induced DNA damage and causes an increase in targeted therapy resistance. sFRP2-induced β-catenin loss promotes invasion and causes increases in ROS, which has been linked to BRAF inhibitor resistance, suggesting that aged patients could benefit from anti-oxidant therapy more than younger patients. The TME is a complicated, unique aspect of cancer which warrants consideration for personalized oncology wherein age, genetics, and tumor mutational load are combined to generate the ideal cancer treatment plan. While much previous research into cancer has focused on genetic modifications and oncogenes, such as KRAS, it has become increasingly clear that a dynamic ECM and TME co-evolving with tumor cells may have a profound effect on proliferation, immune evasion, and metastasis together with the underlying genetic mutations which support tumor initiation. Though tissue-specific aspects of CAF-cancer cell interaction have been reported, other tumor-stroma effects are seen across cancer types and relate to conserved features of a wound-healing response, suggesting a potential avenue for stroma-directed and broadly applicable anticancer therapies.

## 3. A Role for Fibroblast-Derived Metabolites in Tumor-Stroma Interaction

Upon activation, quiescent fibroblasts undergo transcriptomic and metabolic programming that mimics the “Warburg Effect” metabolic phenotype, which is further exaggerated in the hypoxic TME [1,31,32] (Figure 2). The hypoxia response is a universally conserved response to insufficient oxygen availability wherein the hypoxia-inducible factor (HIF) complex is stabilized and transported to hypoxia response element (HRE) promoter sequences to engage in transcription of over 100 genes, many of which are directly involved in increasing anaerobic glycolysis rate [33]. Competition for glucose between CAFs and cancer cells seems counterproductive for tumorigenesis. Instead there are several lines of evidence suggesting metabolic reprogramming to coordinate glucose and lactate metabolism in the TME. Metabolic tracing experiments have shown that well-oxygenated cancer cells support high glycolysis rate of cells in hypoxia by increasing lactate uptake [34,35]. This is further validated by differential expression of monocarboxylate transporters (MCT) with cells experiencing hypoxia increasing MCT4 levels, lactate efflux, and cells in normoxia increasing MCT1 for lactate import [34,35]. In the context of tumor metabolism, this phenomenon has been dubbed as the “Reverse Warburg Effect,” from the perspective of cancer cells reacting to metabolic reprogramming in CAFs [36]. This observation was first made in genetic analyses of breast cancer CAFs that exhibited low caveolin-1 (CAV1) expression [37]. Upon knockout of CAV1, fibroblasts gained myofibroblastic markers and increased rate of aerobic glycolysis [37]. Further analysis demonstrated that the metabolic program of CAFs in breast cancer is supported by cancer cells’ increased lactate uptake to support their own bioenergetic needs [37,38]. The same metabolic reprogramming has been shown in prostate cancer CAFs upon direct contact with cancer cells in co-culture studies [39]. The “Warburg Effect” in CAFs is also exhibited through classic fibroblast activators, TGF-β and PDGF, which have been shown to downregulate isocitrate dehydrogenase (IDH) expression resulting in decreased of cellular levels of α-ketogluterate (α-KG), due to impaired isocitrate to α-KG conversion [40]. α-KG and oxygen are critical co-factors for PHD enzymes, which negatively regulate stability of glycolysis master regulator HIF1α by promoting its ubiquitin-mediated proteosome degradation in normoxia [40,41]. Moreover, TGF-β signaling in CAFs has been shown to trigger increased oxidative stress, autophagy/mitophagy, and aerobic glycolysis—all known factors enhancing HIF1α stability [42,43]. The “Reverse Warburg Effect” does not reflect all metabolic crosstalk across fibrotic tumors as evidenced by studies in breast and pancreatic cancer showing CAFs increasing lactate uptake and re-purposing it for bioenergetics [44,45]. In turn, clearance of lactate by CAFs enhances higher glycolysis rate in cancer cells. It is not clear why well-oxygenated cancer cells will switch to lactate in preference of glucose, but these observations showing the “Reverse Warburg Effect” seem to conform with the concept of metabolic symbiosis required for tumor progression wherein CAFs’ ability to support cancer cell metabolism through glycolysis byproducts is necessary for tumor formation. 

Upon malignant transformation, cancer cells become increasingly dependent on an exogenous supply of amino acids, especially glutamine, the most abundant amino acid in plasma [46]. In ovarian cancer, CAFs have been shown to harness carbon and nitrogen from aspartate, asparagine, and lactate to generate glutamine [47]. Thus, through upregulation of anapleurotic glutamine metabolism CAFs are able to support glutamine-addicted cancer cells in the glutamine-starved TME. This relationship was shown by blocking expression of glutamate ammonia ligase, thereby impairing CAF ability to provide glutamine to cancer cells, which hindered tumor growth [47]. A similar effect was achieved by blocking glutamine catabolism through genetic ablation of glutamate synthase. Combining the effects of inhibiting glutamine synthesis in CAFs and glutamine catabolism in cancer cells prevented formation and metastasis of ovarian cancer. In the nutrient-poor PDAC TME, alanine derived from autophagic CAFs is utilized by pancreatic cancer cells as a carbon source to fuel bioenergetic and biosynthetic processes, compensating for low levels of glucose/glutamine in the TME [44]. Moreover, through the provision of alanine, CAFs further enhance carcinogenesis by allowing cancer cells to fuel TCA cycle, support lipid and NEAA synthesis, as well as diverting glucose metabolism to serine and glycine synthesis both of which are essential for cancer cell survival [44,48]. In contrast to autophagy activation in pancreatic cancer, genetic analyses of prostate and liver tumor stroma shows a decrease of autophagy substrate signaling adaptor protein, p62, resulting in defective autophagy [49,50]. p62 depletion increases expression of ATF4, targeted for ubiquitin-mediated proteasomal degradation in normal fibroblasts with normal expression of p62 [51]. ATF4-positive CAFs support neoplastic cell growth in low glutamine conditions through activation of the pyruvate carboxylase-asparagine synthase pathway [51]. In this instance, CAF-derived asparagine supports cancer cells bioenergetics and nitrogen needs in the glutamine-poor TME. These studies reveal the significance of CAFs as regulators of TME metabolism by providing glutamine and amino acids that serve as intermediates to glutamine metabolic pathways, supporting tumor establishment and metastasis. 

One of the challenges in understanding the role of CAFs in carcinogenesis is the fact that fibroblasts are found in most tissues, yet they remain poorly characterized. Interestingly, a well-known feature of particular pancreatic fibroblasts and hepatic fibroblasts, known as pancreatic and hepatic stellate cells, is their ability to store lipids in their quiescent phenotype [52]. Upon activation and during carcinogenesis, stellate cells lose their ability to store lipids, but the relevance of stromal lipid metabolism in cancer remains poorly understood. Interestingly, metabolomic studies of KRAS mutant cells experiencing hypoxia demonstrate inability to undergo de novo lipogenesis and instead show more reliance on lipid scavenging, suggesting a potential metabolic function of lipids secreted by activated fibroblasts [53]. Recently, work from our lab demonstrated that activated stellate cells secrete abundant lysophosphatidylcholines (LPC), the preferred fatty acid scavenging substrate for RAS-transformed cells, which can support PDAC cells growth both via uptake and biomass production and via hydrolysis by the secreted enzyme autotaxin to yield mitogenic lysophosphatidic acid (LPA) [54]. Reprogramming of lipid metabolism has also been demonstrated in prostate CAFs compared to normal prostate fibroblasts [55]. Prostate CAFs have elevated neutral lipid storage, and these lipid stores cooperate with pigment epithelium-derived growth factor to amplify microtubule-organizing centers. Further roles of CAF lipid secretion remain to be elucidated as several lines of evidence suggest that exogenous lipids support auxotrophic cancer cell growth [56,57]. 

Another little-explored aspect of CAFs is exosome-mediated metabolic crosstalk. Recent studies have shown that exosomes carry proteins, nucleic acids, miRNAs, and metabolic molecules [58]. CAF-derived exosomes, CDEs, of prostate and pancreatic cancer have been shown to reprogram metabolism of cancer cells by significantly up-regulating glycolysis while down-regulating oxidative metabolism by promoting glutamine decarboxylation and at the same time generating metabolites for de novo lipogenesis [59]. Metabolomic analysis of CDE contents reveal that they carry lactate, acetate, and amino acids—shown to be taken by cancer cells through carbon tracing analyses. Interestingly, CDE metabolic reprogramming was shown to be independent of oncogenic KRAS in pancreatic cancer, thereby demonstrating CAFs’ ability to reprogram and support cancer cell metabolism independent of oncogene activation. CDEs have been shown to enhance gemcitabine resistance in pancreatic cancer cells by enhancing proliferation and glycolysis [60]. The extent to which CDEs modulate the metabolism of cells in the TME remains to be further explored.

The TME is highly dynamic and the accessibility to oxygen and nutrients is never constant forcing cancer cells to reprogram their metabolism accordingly. The studies summarized in this section highlight the role CAFs play in sustaining tumor cell metabolism. Further studies are required to understand the way in which CAFs provide metabolic support to cancer cells in order to identify novel therapeutic avenues. 

## 4. Fibroblasts as Determinants of Systemic Metabolism in Cancer

Beyond metabolic dysregulation in its local tissue context, cancer is associated with metabolic alterations in the host [61]. Abnormal whole-body metabolic responses to cancer include cancer cachexia, a potentially lethal wasting syndrome driven by negative energy balance and associated with loss of adipose and muscle tissue [62]. Cachexia has been mechanistically linked to the inflammatory response to cancer, and particularly to elevated levels of systemic pro-inflammatory cytokines [63]. Cachexia is a common and early event in the pathogenesis of some cancer types, and evidence of tissue breakdown associated with cachexia may even be a biomarker of early tumorigenesis [64,65]. Though mechanisms driving cancer cachexia are complex and remain to be elucidated, early evidence has emerged that CAFs may play a role in tissue wasting, in part by mediating an inflammatory response and in part through direct interactions with relevant host tissues. Fibroblast activation protein-α (FAPα) marks activated fibroblastic cells in tumors [66] and other pathologic inflammatory conditions, including atherosclerosis [67]. FAPα-positive cells in the primary tumor microenvironment have been associated with immune suppression, promoting T cell exclusion via secretion of CXCL12 [68]. However, recent work using a FAPα reporter in mice showed that these FAPα-expressing fibroblastic cells can be found in numerous tissues in the adult mouse, including skeletal muscle [69]. FAPα-positive cells across tissue contexts have similar transcriptomes, suggesting a common lineage. Depletion of FAPα-positive cells in healthy mice caused a cachexia-like syndrome, characterized by rapid weight loss and reduced muscle mass despite adequate food intake. FAPα-positive fibroblasts in skeletal muscle were shown to be the predominant source of Lama2 and Follistatin (Fst288 and Fst315), key regulators of myofiber thickness and muscle growth. Thus, loss of FAPα-positive fibroblasts from skeletal muscle was proposed to play a causal role in the muscle-wasting aspect of cachexia. Strikingly, the authors observed significant loss of FAPα-positive cells from skeletal muscle in cachectic tumor models. These findings implicate fibroblastic cells in maintenance of muscle mass, and raise the possibility that fibroblast loss from skeletal muscle promotes the muscle wasting observed in cancer cachexia, with major implications for systemic metabolism.

The host metabolic perturbations associated with cancer progression have been partly attributed to increased systemic levels of pro-inflammatory cytokines [63]. IL6 in particular has been functionally linked to cachexia [70,71,72,73,74,75,76,77], and is elevated in patients with cachexia-associated cancers [78,79,80]. Further, activation of STAT3 downstream of IL6 has been linked to muscle wasting in cancer [81]. In multiple cancer types, CAFs are reported as a significant source of IL6 in the tumor microenvironment [7,82,83,84,85], highlighting CAF-derived IL6 as a potential link to cancer cachexia. Shining light on the role of IL6 in cancer cachexia, recent work demonstrated that the elevated IL6 in cachexia-associated tumor models suppresses hepatic ketogenesis, by downregulating expression of master ketogenic regulator PPARα in the liver [70]. IL6 promoted metabolic stress in response to caloric restriction, including elevated corticosterone levels, and recombinant IL6 lowered fasting ketone and glucose levels. Interestingly, the increase in systemic glucocorticoids in response to IL6-mediated suppression of hepatic ketogenesis was associated with a suppression of anti-tumor immunity. Reduced food intake was a driver of the increase in glucocorticoids and immune suppression, and caloric deficiency is commonly seen among patients with cachexia-associated cancers. Notably, IL6 can also directly regulate the hypothalamic-pituitary-adrenal axis [86], and may further contribute to cachexia through its activity in the brain. While CAFs can function to locally suppress anti-tumor immunity, these findings raise the possibility that CAFs participate in a complex metabolic and inflammatory host response, leading to systemic elevation of glucocorticoids and immune suppression. The role of IL6 specifically derived from CAFs has yet to be tested in this axis. 

While studies of the metabolic, immune-modulatory, or paracrine signaling functions of CAFs in various solid tumors suggest tumor-supportive roles for these cells, three papers published in 2014 demonstrated a protective role for CAFs in pancreatic cancer with respect to survival outcome [87,88,89]. To probe the roles of the abundant CAF population in these tumors, the authors used genetic or pharmacologic approaches to ablate CAFs during pancreatic tumorigenesis, either ablating Shh-dependent CAFs [87,89] or αSMA-positive CAFs [88]. Though these different systems yielded somewhat different results, these studies together provide compelling evidence that CAF ablation causes mice to succumb significantly earlier to the disease compared to CAF-replete controls. Interestingly, in the study by Rhim et al. employing both genetic and pharmacologic inhibition of Shh to ablate CAFs, the authors report that mice succumb with very small tumors, but with severe cachexia, including wasting of adipose tissue and muscle exceeding that seen in controls. Data are shown from systemic Shh inhibition, but the authors report that the same phenotype was observed in their genetic model which specifically targets Shh in the pancreas, and therefore specifically inhibits Shh-dependent CAFs within the local tumor microenvironment. This raises the intriguing possibility that pancreatic CAFs promote improved survival outcomes in part by inhibiting pro-cachectic mechanisms within the primary tumor. A mechanistic connection between Shh-dependent CAF function and critical mediators of cancer cachexia has not been established. However, further investigation into this axis may be warranted, as therapies targeting pancreatic CAFs would ideally leave any such cachexia-suppressive mechanisms intact.

Likely related to their evolutionary role in the wound-healing response, CAFs are important sources of growth factors in the tumor microenvironment, as discussed above. In multiple solid tumor types, CAFs have been described as significant sources of growth factor ligands for the epidermal growth factor receptor (EGFR), including high-affinity ligands betacellulin (BTC) [90] and heparin-binding EGF-like growth factor (HB-EGF) [91] as well as lower-affinity ligand epiregulin (EREG) [92]. CAFs and normal fibroblasts are also prominent producers of parathyroid hormone-related protein (PTHrP) [93,94], a developmental regulatory molecule activated by EGFR signaling [95]. A recent study aimed to identify novel regulators of cancer cachexia, and found a novel connection between factors that promote adipose tissue browning and the onset of features of cachexia including weight loss and muscle atrophy [96]. Cachexia is characterized in part by increased resting energy expenditure, which has been linked to increased thermogenesis by brown adipose tissue [97,98,99,100] and to browning of white adipose tissue [101]. Kir et al. found that Lewis lung carcinoma (LLC) cells induce adipose tissue browning and cachexia. By comparing gene expression in more thermogenic versus less thermogenic clones, they identified candidate paracrine thermogenic regulators. By testing candidate recombinant proteins, the authors found that the 3 EGFR ligands discussed above—BTC, HB-EGF, and EREG—as well as PTHrP all stimulate thermogenic gene expression in primary adipocytes. Though the study focused on cancer cell-derived PTHrP as a key regulator of adipose tissue browning in the LLC system, a link between EGFR ligand production and cancer cachexia is intriguing and warrants further study. Activation of EGFR/MEK signaling in the primary tumor has been recently linked to MEK activation and wasting in host tissues [102], and while CAFs as a source for these ligands or for PTHrP have not been specifically addressed, EGFR signaling has been functionally linked to cachexia and/or energy expenditure in additional systems [103,104]. Providing ligands which act either via tumor cells or directly on adipocytes to promote thermogenesis might suggest a deleterious role for CAFs as promoters of cancer cachexia, and such a role may indeed be tissue- and context-dependent.

## 5. Conclusions and Future Directions

Non-malignant cells of the tumor microenvironment, including but not limited to CAFs, exert an important influence on key metabolic pathways in cancer cells and on intratumoral metabolite levels. The significance of these paracrine interactions warrants further study in vivo, as cancer cells exhibit specific and complex metabolic requirements within host tissues [105,106,107,108,109] that are difficult to model using in vitro systems. CAFs present a limitation in this regard, as specific Cre lines to achieve genetic manipulation in these cells are presently lacking. Further, while studies of metabolite exchange in vitro have established important modes of cell-cell contact within the tumor microenvironment, validation and further investigation of these interactions will be bolstered by emerging means to study intercellular metabolic relationships within tissues [15]. In considering CAF-cancer cell interactions as potential therapeutic targets, it will be important to understand the critical and non-redundant metabolic functions of CAFs that enable cancer cells to maintain their proliferative capacity within a nutrient-poor tumor microenvironment. As cancer cells exhibit metabolic plasticity [110,111], therapies targeting metabolism-modulating pathways will likely need to target parallel mechanisms fulfilling bioenergetic needs, or to combine metabolic inhibitors with therapeutic interventions that suppress cancer cell plasticity and thus the capacity for metabolic adaptation. Further, as suppression of anti-tumor immunity is increasingly linked to intratumoral metabolite levels and to activity of key metabolic pathways in immune cells [112], the consequence of the CAF secretome on the metabolism and function of immune cells in the tumor microenvironment warrants investigation. 

## Figures and Tables

**Figure 1 cancers-11-00619-f001:**
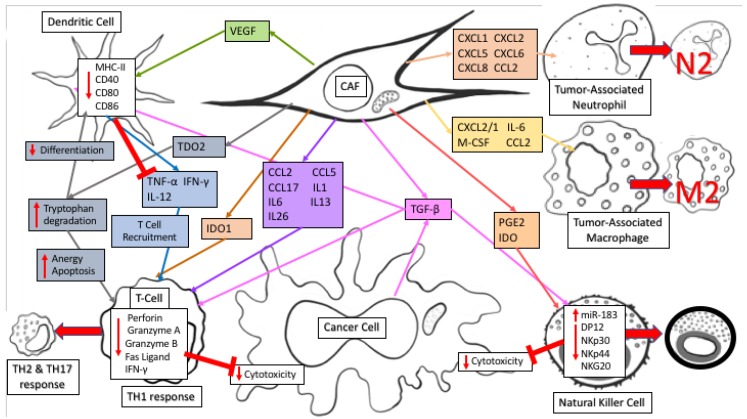
CAF secretions of immunomodulatory factors regulate anti-tumor immune response. CAF secretion of CXCL12/SDF1, M-CSF/CSF-1, IL-6, and CCL2/MCP-1 recruits tumor-associated macrophages to the TME and helps differentiate them to an M2 Immunosuppressive phenotype. CXCL1, CXCL2, CXCL5, CXCL6, CXCL8, and CCL2 recruits tumor-associated neutrophils and polarizes them to an N2 pro-tumoral phenotype. CAF and cancer cell secretion of TGF-β induces miR-183 to inhibit DAP12 transcription and reduce natural killer (NK) activating receptors (NKp30, NKp44, NKG2D), while PGE2 and IDO secretion decreases NK cell cytotoxicity against cancer cells. TGF-β also promotes cell death of CD8+ T cells by inhibiting expression of Bcl-2 and causes dendritic cells (DC) to downregulate expression of MHC class II, CD40, CD80, and CD86 leading to decreased antigen presentation efficiency along with decreased production of TNF-α, IFN-γ, and IL-12. TDO2 and IDO1 secretion promotes tryptophan metabolism to kynurenines (Kyn), inhibiting DC differentiation and damages T cell response by catabolizing tryptophan degradation into Kyn, causing T cell anergy and apoptosis through depletion of tryptophan and accumulation of immunosuppressive tryptophan catabolites. VEGF secretion inhibits DC generation and maturation by reducing MHC class II expression and antigen presenting abilities. Secretion of CCL2, CCL5, and CCL17 along with polarizing cytokines IL-1, IL-6, IL-13, and IL-26 switch CD4+ helper T lymphocytes from an anti-tumor T_H_1 response to a pro-tumor T_H_2 and T_H_17 response.

**Figure 2 cancers-11-00619-f002:**
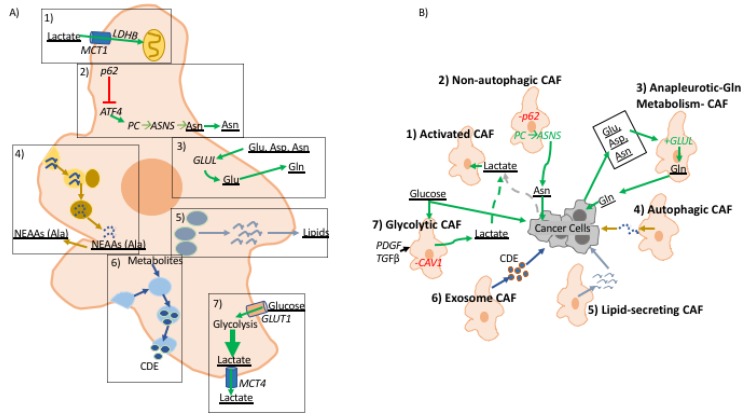
A role for fibroblast-derived metabolites in tumor-stroma interactions. (**A**) illustrates metabolic reprogramming in CAFs and (**B**) illustrates the effect of metabolites derived from CAFs to cancer cells in the TME. (1) Activated CAFs uptake excess TME lactate produced by glycolytic cells in the TME. (2) p62 deficient CAFs are autophagy defective and have upregulation of ATF4 which activates metabolic flux through pyruvate carboxylase (PC) → asparagine synthase (ASNS) pathway producing asparagine that is consumed by cancer cells. (3) CAFs characterized by upregulation of anapleurotic glutamine metabolism where cancer cell-derived aspartate, asparagine, and glutamate are used to generate glutamate that is fed to cancer cells; glutamine amino ligase (GLUL) is overexpressed in these CAFs. (4) Cancer cells induce autophagy in CAFs increasing turnover of non-essential amino acids, alanine has been shown to be up-taken by cancer cells to support growth. (5) Upon activation pancreatic and hepatic CAFs shift from lipid storing to lipid-secreting wherein CAF-derived lipids support proliferation and migratory potential of cancer cells. (6) Exosome releasing CAFs pack metabolic molecules used by cancer cells. (7) CAV1 deficient CAFs are known to upregulate glycolytic metabolism, as well as, fibroblast activation by PDGF and TGFβ. (**A**) describes metabolic alterations in CAFs in dashed boxes. In both figures metabolic pathways are denoted with green arrows, cellular processes are assigned specific colors, and metabolic molecules are underlined.

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
