# Peer review of "Fibroblasts as Modulators of Local and Systemic Cancer Metabolism"

_cancers, 2019, doi:10.3390/cancers11050619_

Round 1
Reviewer 1 Report
This review by Sandford-Crane et al entitled "Fibroblasts as modulators of local and systemic cancer metabolism" summarizes the metabolic function of CAF that might be associated with their pro-tumorigenic behavior and gives insights into the potential players responsible for this effect. Overall is a well-presented and comprehensive review. There are a few comments/suggestions that should be included:
1) The figures are very informative, however the font size chosen is small and hard to read. Also the intensity of the background in some of the boxes are too dark. Please use lighter tones
2) In regards to the role of lipid storage in fibroblasts fueling cancer cell behavior. A recent paper by Nardi et al in J Cell Sci 2018 (DOI: 10.1242/jcs.213579) suggest that CAF can also re-program lipid metabolism and affect MTOC biology to support tumor growth. Perhaps a line of tow of discussion will help the readers get some information of the work in this area.
Author Response
We thank the reviewer for these suggestions. Font sizes and colors in the figures have been edited as suggested, and the J Cell Sci paper has been added to the manuscript.
Reviewer 2 Report
This is an informative review on the role of fibroblasts in cancer progression, focusing on regulation of cancer cell metabolism as well as systemic metabolism. I have only a few minor comments to improve this manuscript.
1) line 108-109, 137-139, “TGF-b promotes cell death of CD8+ T cells by inhibiting expression of genes involved in cytotoxic function…”: This sentence is misleading. Ref. 27 describes that “TGF-b promotes cell death of effector CD8+ cells by inhibiting expression of the pro-survival protein Bcl-2.” Inhibition of genes involved in cytotoxic function does not appear to be the cause of cell death.
2) Figure 1: Letters and arrowheads are too small. In addition, there should be an arrow between TGF-beta and T-cells. Is the arrow from IFN-g to CAF explained in the legend?
3) Figure 2: Metabolites are underlined in A, while cell types in TME are underlined in B. This is somewhat confusing and should be explained in the legend. Again, letters are too small.
4) line 194 negtively >> negatively
Author Response
We thank the reviewer for these helpful comments. The TGF-b explanation has been corrected, the suggested edits have been made to both figures, and the spelling error has been fixed.